# Managing Innovation Resources in Accordance with Sustainable Development Ethics: Typological Analysis

**Anastasia O. Ljovkina** [1], **David L. Dusseault** [2], **Olga V. Zaharova** [3] **and Yury Klochkov** [4,*]

1   Department of Economic Security, Tyumen State University, Tyumen 625003, Russia;
    anastasia.ljovkina@gmail.com
2   Department of Philosophy, Tyumen State University, Tyumen 625003, Russia; d.dusseault@utmn.ru
3   School of Advanced Studies, Tyumen State University, Tyumen 625003, Russia; o.v.zakharova@utmn.ru
4   Monitoring Center of Science and Education, Peter the Great St. Petersburg Polytechnic University,
    St. Petersburg 195251, Russia
*   Correspondence: y.kloch@gmail.com; Tel.: +7-927-184-90-01

**Abstract:** The regional development path depends on managing innovation resources. However, increasing the quantity of innovation activity and managing innovation resources only by financial indicators does not guarantee progress in sustainable development. This paper argues that basic conditions for effective sustainability-oriented innovation activities are: (1) the accordance of relevant activities with sustainable development ethics and (2) their marked focus on systemic and long-term sustainable development targets. These parameters can be considered fundamental principles for designing and developing effective sustainability-oriented innovation systems and innovation policies. Analysis of the two basic principles precedes estimation of the effectiveness of innovation activities, innovation systems, or innovation policies. In this paper, a special typological analysis technique was applied to assess basic conditions for the effectiveness of sustainability-oriented innovation activities observed in the case of the Tyumen region, Russia. It was found that since 2009 the Tyumen regional innovation support system has not been conceptually designed in accordance with sustainable development ethics or considering a long-term vision. Therefore, the projects themselves afford only temporary solutions to regional problems by implementing innovations that mainly have short-term and mid-term social–economic effects. As a result of the analysis of fundamental conditions for sustainability-oriented innovation activities in Tyumen region, this paper proposes recommendations on necessary measures for redesigning decision-making principles of regional innovation support systems in order to significantly increase the potential impact on the development of a truly sustainable regional economy.

**Keywords:** innovation resources; innovation support system; resources typological analysis; sustainable development; sustainability-oriented innovations

## 1. Introduction

### 1.1. Sustainability Crisis

Outcomes of contemporary world social–economic development are very ambiguous and raise many questions. On the one hand, in the 1970s, society admitted the limits of economic growth [1], rethought the destructive consequences of consumerist culture [2,3], and discussed the necessity of replacing capital-oriented and "hooked on growth" economic policies with ones which foster sustainable forms of economic behavior [4–6]. In 1983, the UN General Assembly Universal Sustainable Development Agenda defined what developing and implementing national sustainable development

(SD) programs entail globally [7–9]. On the other hand, symptoms of deep global crisis—global social, economic, and environmental problems—continue to increase their presence [10,11], and as such demonstrate that humanity has continued along the "business as usual" trajectory [12,13]. Such trends indicate an overall lack of urgency regarding the need for drastic changes, both in institutions and policies [14]. Hence, the risk of facing catastrophic economic, ecological, political, and social consequences passing the "point of no return", as predicted in a previous study [1], without strategic and systemic changes is very high. As was noticed in the latest report of the Rome Club, the tactic of accelerating growth and trade has led to far less socio-economic progress than is really required, and as such a radical new synthesis is needed [10].

### 1.2. Focusing on Long-Term Rationality

Based on the evidence above, there is a crucial need for strategic changes, i.e., strategic innovations (social–economic, technological, and educational) that may help to alleviate the deep causes of global problems. According to Raworth, these causes are growth-addicted trickle-down economic policies, self-contained markets divorced from the sphere of social life, prevailing individualism, short-termism, the predominantly mechanical approach to managing contemporary social–economic complexity, and, in general, a lack of planetary vision for an effective economy [15].

Significant cultural changes and fundamental social transformations are necessary to overcome the "moral collapse" of today's world order [16,17], in addition to shifting towards more sustainable modes of production and consumption [18,19]. Thus, more systemic and strategic changes are required, such as "changes to current economic systems and priorities that are largely beyond the scope of a low carbon industrial strategy" [20] (p. 19) or a particular non-systemic sustainable development strategy in general.

In other words, a shift from an unsustainable to a sustainable society requires radical structural changes, providing a "Deep Transition" [21] through the series of connected transitions to sustainable development in many socio-technical systems [22–25]. According to the Human and Nature Dynamics (HANDY) model, avoiding civilizational collapse requires major policy changes, including a significant decrease in the depletion of nature and a major reduction in inequality [26], which in turn require appropriate strategic social, economic, and environmental innovation. If implemented efficiently and under suitable conditions, strategic social innovation activities have the potential to transform social institutions and mechanisms to match SD goals [27] and SD ethics [12]. SD ethics, or in other words, ethical foundations of SD, followed from the definition given by United Nations Commission on Environment and Development—"development that meets the needs of the present without compromising the ability of future generations to meet their own needs" [7]. It implies not only SD triple dimensions but also outlines the necessity of considering long-term SD effects (including effects of innovation activities) [12] (pp. 19–23). Strategic technological innovation activities are, thus, crucial for solving critical resource and environmental problems plaguing today's society, including the development of environmentally clean, safe, and efficient sources of energy, and waste-free and zero-emission production technologies [28–31].

Thus, systemic and strategic innovations tailored to solving critical issues of sustainable development (SD) from a long-term perspective are key factors for the deep SD societal transition and should be prioritized in today's innovation activities [32,33]. Subsequently, innovation systems must be fit for purpose; that is, in accordance with (1) SD ethics and (2) long-term rationality or significant strategic and systemic effects [34–37]. These conditions should be considered as basic principles for designing and developing effective sustainability-oriented innovation systems and innovation policies [38,39]. Applying financial indicators and benchmarking methods to estimate the effectiveness of sustainability-oriented innovation activities and systems makes sense only after the analysis of their concordance with the previously mentioned basic principles of sustainability-oriented systems.

### 1.3. Previous Research

Innovation policy discussions, according to Schot and Steinmueller, exist in three dominant established frames. First, focusing on the institutionalization of government support for science and research and development aimed at economic growth. Second, an emphasis on competitiveness, which is shaped by national systems of innovation for knowledge creation and commercialization. Third, the call for SD transformative change due to contemporary social and environmental challenges [40]. The authors argue "all three frames are relevant for policymaking, but exploring options for transformative innovation policy should be a priority" [40].

After the term "sustainable development" was first coined at the United Nations Conference on the Human Environment in 1972 and the Sustainable Development Agenda was manifested by the UN General Assembly in 1983, scientists' and politicians' attention turned to the importance of environmentally sound technologies and innovation aimed at sustainability [41,42]. Issues concerning technology, innovation, and management for sustainable growth have been extensively studied from different perspectives. However, there is still a great demand for methodological studies that investigate sustainability-oriented innovation activities from a more holistic perspective [43] and from the point of view of their transformative and transitive potential [21,44,45].

Our preceding studies resulted in discovering a lack of a general methodological discourse in terms of assessing innovation activities within the frame of SD transformation. The absence of analytical methods assessing basic conditions for effective sustainability-oriented innovation systems was also noted [46]. The universally recognized guidelines laid out in the "Oslo manual" (2005) provide a methodology for collecting data about innovation activity, but it does not include classifications of innovation specifically in tune with SD goals [47].

The Oslo manual configures characteristics of data sets, which are used according to various methods of estimating innovation activities and system effectiveness. Some of the popular benchmarking methods related to national and regional innovation system effectiveness include indicators that are well-adjusted with SD goals. These methods are the Science, Technology, and Industry Scoreboard (STI Scoreboard), Sustainability-adjusted Global Competitiveness Index (GCI), and the Technological Barometer [48]. However, they still do not take into account long-term rationality of innovation activities—the scale and depth of a set of innovations' impacts on solving sustainability problems. Still, other benchmarking methods only partially allow for estimating innovation activities and innovation systems in accordance with general SD goals. For example, the Innovation Union Scoreboard, Regional Innovation Scoreboard, and Innobarometer include indicators showing the extent of social progress, but do not consider an environmental perspective.

Technological innovations cannot provide deep transition to a sustainable world by emerging spontaneously or by being implemented at their own pace [49] under market laws. They require systemic and systematic understanding, assessment, planning, management, and monitoring of "the effectiveness of strategies, policy, or practices from a holistic perspective" [50] and political guidance to orchestrate ecological modernization and support of a SD societal transition [51,52]. A basic lack of harmony between the implementation of innovation systems with SD goals and their weak focus on strategic innovation predetermines a low impact on SD transformation, even if their effectiveness as measured by the standard techniques is quite high. Thus, there is an urgent need for a radical increase in the effectiveness of sustainability-oriented innovation activities. Recognizing the lack of theory and experience, which provide appropriate conceptual analysis and act as a guide to policymaking, in this research, we aim to develop and test an instrument to analyze the basic conditions for effective sustainability-oriented innovation activities.

### 1.4. Research Outline

The practical purpose of this research is to provide respective audiences with a relevant tool for assessment and conceptual design of innovation activities and innovation systems tuned to provide

significant SD social transformations. To do so, we developed an original matrix typology and classification of identifiable innovations.

In order to test our analytical method and illustrate its practical application, we performed a case study centered upon innovation activities in the Tyumen region, Russia. We created a typological analysis of all innovation projects supported by the local government since 2009. Finally, on the basis of this analysis, we made recommendations for redesigning the innovation support policy to significantly increase the impact of regional innovation activities to develop a truly sustainable regional economy. We consider the suggested method as a management instrument that provides common ground for matching interests of different social agents—public administration, the corporate sector, society, and the scientific community. Thus, we see great potential in its application for decision making in the innovation sphere.

This paper is organized as follows. Section 2 describes the methodological framework used to design the typological analysis and also describes research materials and methods. Section 3 presents the results of the typological analysis of supported innovation projects and we discuss the results in Section 4. Finally, Section 5 presents our conclusions, including recommendation, research implications, limitations, and further research development perspectives.

## 2. Methodology

### 2.1. Conceptual Design of Typological Analysis.

Noticeable shifts in ethical norms of social–economic activities began with a growing understanding of the scale of global problems, in particular, environmental ones [1]. In 1972, the United Nations Environment Programme was launched and in 1987 the term "sustainable development" was first presented in the report "Our common future" by the World Commission on Environment and Development. The UN Environment Programme and the following international SD agreements have subsequently developed into the national, regional, and local SD programs all over the world.

Innovation activity is generally cited as the main driver of today's societal transformations. Thus, to bring us to a future of common prosperity, innovation activities should match SD ethics and prioritize implementation of those innovations with the potential to provide the greatest SD impacts, rather than by the greatest investment returns. The role of science and innovation was also reconsidered in UNESCO's *Declaration on Science and the Use of Scientific Knowledge*. It states, "The practice of scientific research and the use of knowledge from that research should always focus on the welfare of humankind, including the reduction of poverty, should be respectful of the dignity and rights of human beings, and of the global environment, and fully consider our responsibility towards present and future generations. There should be a new commitment to these important principles by all parties concerned" [53].

To analyze the basic conditions for effective sustainability-oriented innovation activities, a special typological analytical technique was developed. To classify innovations according to their SD effects, we suggested two typological dimensions: (1) the accordance of innovation activities with SD ethics and (2) the scale of SD innovation effects.

(1) *SD ethics* implies the triple bottom lines of SD: Economic, Social, and Environmental factors [12,54].

*The Environmental Dimension* implies providing healthy integration of biological ecosystems and the physical environment (including the human environment) and improving inherent abilities for self-recovery. This dimension includes both social innovations that change human behavior towards the environment in general and specific eco-innovations providing a more rational use of natural resources, thus reducing society's ecological footprint. According to Kemp and Pearson, eco-innovations imply "the production, assimilation, or exploitation of a product, production process, service, or management or business method that is novel to the organization (developing or adopting it), and which results,

throughout its life cycle, in a reduction of environmental risk, pollution, and other negative impacts of resource use (including energy use) compared to relevant alternatives" [55] (p. 7).

*The Social Dimension* assumes a peaceful world and social progress through reaching prosperity and ensuring equal opportunities for all, increasing possibilities for human creativeness and self-realization. Innovative effects in the Social Dimension involve the creation of constructive conditions for personal, social, and professional development, the absence of destructive conflicts among people, and the development of collaborative methods to solve common problems based on humanistic values (such as a value of human life, freedom, and development). Any technological innovation considering SD environmental and economic dimensions cannot avert civilizational collapse without fundamental changes in society's dominant values of growth, exploitation, and consumption [56].

*The Economic Dimension* suggests achieving common prosperity through steady-state economic development based on the principles of rational consumption, clean and effective production, and problem-oriented innovation development [57] (p. 1). In particular, the effects of innovations within the SD economic dimension imply developing resource and energy-saving technological innovations, effective recycling technologies, longer lifetime products, and social innovations, leading to a more sustainable economy.

*(2)* The *scale of SD innovation effects* can be characterized by the level and complexity of innovations' SD effects, the quality of global problem solutions, the extent of influence on problem situations, and the time frames of effects (Table 1). According to the "scale" dimension, we classify innovations into three conditional categories: (a) strategic (eradicating deep systemic causes of the greatest common problems of humanity and providing radical SD societal transition), (b) tactical (optimizing social–economic life and environmental situations within the current cultural context), and (c) operative (aimed at alleviating the consequences or "symptoms" of systemic global problems).

**Table 1.** Classification of innovations by the scale of sustainable development effects.

| Typological Characteristics | Types of Innovations by the Scale of SD Effects | | |
|---|---|---|---|
| | **Strategic** | **Tactical** | **Operative** |
| The level and complexity of innovations' effects | System transformative effects | Some integrative effects | Immediate single effects |
| The quality of problem solution | Eradicating the radical SD problems through eliminating their causes | Preventing some consequences of SD problems (or derivative SD problems) | Alleviating consequences of SD problems |
| The extent of influencing the problem situation | Radical societal transition to SD path | Noticeable improvement of the global situation in SD context | Insignificant surface improvements |
| Time frames of effects | Long-term | Mid-term | Short-term |

The overall typological model considers both SD ethics dimension and a scale of innovative SD effects. The model for analysis of basic conditions for the effective sustainability-oriented innovation activities can be observed in matrix view (Table 2).

**Table 2.** Typological matrix for classification of sustainability-oriented innovation activities.

| The scale of SD Innovation Effects | SD Ethics Dimensions | | |
|---|---|---|---|
| | **Environmental** | **Social** | **Economic** |
| Strategic | Strategic environmental innovations (SEnI) | Strategic social innovations (SSI) | Strategic economic innovations (SEI) |
| Tactical | Tactical environmental innovations (TEnI) | Tactical social innovations (TSI) | Tactical economic innovations (TEI) |
| Operative | Operative environmental innovations (OEnI) | Operative social innovations (OSI) | Operative economic innovations (OEI) |

According to the typological matrix, we end up with nine innovations classification groups that are defined at the intersection of their ethical and scale dimensions:

- *Strategic environmental innovations (SEnI)* are aimed at eradicating deep systemic causes of global environmental problems through the implementation and expansion of radical environmentally-oriented innovative technologies. Examples include: innovations in the sphere of clean energy, "free" or renewable energy sources, clean production technologies, non-waste production, and consumption cycles.

- *Strategic social innovations (SSI)* are intended to eliminate identifiable causes of global social problems through the implementation of radically innovative social technologies, thus transforming the basis of current social–economic systems, interactions, and relationships, and providing significant, steady humanization of societal systems. Examples include: innovations that increase the role of every human in social life and development, establishing priorities for individual and social collaborative development over money, profit, and competition; holistic systems of health-saving technologies; readily available, effective medicine, eliminating the causes of human illnesses and substantially improving human health, immune systems, life quality, and life expectancy; social, psychological, and educational innovations, which help people develop an active personality with free will, an extended mind, critical, logical, and systemic thinking, and the ability to find deep cause–effect relationships and to predict long-term effects; and social technologies that support innovators and innovations with high social significance.

- *Strategic economic innovations (SEI)* serve the purpose of solving systemic global economic problems, such as poverty, resource deficits, over-production, and over-consumption, and eliminate economic reasons for wars and catastrophe. SEIs improve the life quality of the whole society, recondition economic relationships between humans based on the premise that "the economic" serves "the social", and not vice versa. The goals and means of SEI match the basic humanistic values and principles of a Steady-State Economy and their results significantly increase the integrated ecological, social, and economic wealth of society. Examples include: local currencies, collaborative and sharing economies, blockchain technology, and automation of unsafe and unhealthy production operations.

- *Tactical environmental innovations (TEI)* are designed to prevent the worst consequences of radical environmental problems and solve particular environmental problems without touching their deep causes or anthropogenic character. Examples include: innovative technologies for resource-saving production in particular industries; technologies that significantly decrease environmental pollution and environmental threats in mining industries and other unsafe production, and new methods and reagents for cleaning seas and oceans from oil and other pollution.

- *Tactical social innovations (TSI)* are aimed at preventing and solving significant social problems without radically eliminating their root causes. Examples include: new methods for developing tactical analytical thinking in the educational system; increasing the level of automation and computerization of manual and intellectual labor in existing workplaces, improving information and communication technologies and infrastructure; implementing effective mechanisms and systems supporting innovators who develop market-oriented and profitable innovations; developing health saving technologies, and curing and revitalizing medical technologies.

- *Tactical economic innovations (TEI)* provide significant mid-term economic SD effects. They are aimed at solving economic problems at the tactical level using innovative methods and instruments that are available in the framework of the current socio-economic system. Results of implementation of TEI slow down resource exhaustion, ease environmental problems, improve food safety, and increase economic prosperity. Examples include: innovations providing competitiveness of a particular territory; in particular, technologies that decrease resource consumption and increase availability of material welfare.

- *Operative environmental innovations (OEnI)* deal with the particular multi-layered implications of consequences associated with global climate change. They are new technologies or practices aimed at preserving threatened species, innovative methods of eliminating consequences of anthropogenic and natural catastrophes, along with innovations that help people adapt to unfavorable ecological

conditions. Examples include: new technologies for cleaning water after oil spills, new materials for filtering potable water, and new technologies for building houses from plastic waste.

- *Operative social innovations (OSI)* are aimed at alleviating the most urgent social problems, which can be characterized as consequences of root social problems (non-effective and conflict mechanisms of the actual social–economic system). OSIs provide step-by-step improvements in social institutions and organizations within the framework of existing policies and processes. OSIs are mainly oriented towards profitable and short-term health policies and programs. Examples include: optimization of existing bureaucratic procedures; new medicines that cure symptoms more effectively or provide better pain relief; new ways of identifying people who need urgent psychological or financial aid; new psychological techniques that help people to overcome stress because of unemployment, gender, ethnicity, or income inequality.

- *Operative economic innovations (OEI)* imply new ways to create short-term profit maximization. Examples include: innovations that increase the effectiveness and scale of mining industries and existing forms of production; new technologies that increase the possibility of extract finite, non-renewable resources; and marketing innovations that boost consumption.

### *2.2. Materials and Methods*

In our adopted case study, the basic conditions necessary to foster effective sustainability-oriented innovation activities were assessed in the Tyumen region. The region is one of the largest and the richest in natural resources (especially oil and gas) among Russian regions. It possesses a rather favorable economic and geographical position due to its location between the Arctic Ocean and the southern state border between Russia and Kazakhstan, the region's proximity to economically-developed regions of Russia located west of the Urals, and the abundance of various natural resources, in particular vast reserves of oil and natural gas. Included within the territorial and administrative structures of the Tyumen region are two autonomous districts: Khanty-Mansi and Yamalo-Nenets.

From a policy perspective, the region's authorities have made complex yet sustainable development a core strategic goal for the local economy, as declared previously [58]. In particular, innovation activity is identified as a key factor of regional sustainable development and is seen as a major driver for stimulating and supporting innovation activities in cooperation with the Tyumen government.

For its part, the West-Siberian Innovation Center (Tyumen Technopark) is the main institute for regional innovation support infrastructure. It was founded by the Department of Investment Policy and Government Support of Entrepreneurship on July 7, 2009. The Technopark supplies complex support for regional innovation activities at all stages of project development, beginning from the conceptual design stage and ending with the application of innovative technologies in regional manufacturing or the service economy. The accordance of supported innovation activities with basic principles for designing and developing effective sustainability-oriented innovation systems is the main factor for successful regional sustainable development [59–66]. Therefore, overall sustainability for the region largely depends on the Technopark decision-making policy regarding which particular innovations will be supported by the center and which will not. To determine the degree to which the Technopark was effective at supporting regional sustainability goals, the Technopark's project results (supported innovations) were assessed according to the basic principles for effective sustainability-oriented innovation activities, namely, (1) the suitability of supported innovation activities with SD ethics and (2) the scale of SD innovation effects.

Technopark's innovation activity gradually grew over the first 5–6 years. For this period, there was a separate database of supported innovation projects for this period. Primary data was collected from official open internet sources: (1) the Tyumen regional government official web-page [67], containing the list of supported innovation projects during 2009–2015 and additional project descriptions on the Internet (31projects in total); and (2) the Technopark's official web-page [68], containing the list of supported innovation projects during 2016–2017 (66 projects in total). We did not include projects supported in 2018, as there were no full descriptions for them at the time of this research.

In our calculations, we considered innovation projects in the year of the Technopark's decision to support them. For example, if the project was supported in 2015 it was considered only in this year (2015), despite its further development in 2016 and later.

We obtained the full descriptions of supported innovation projects from the web pages of Technopark's residents and used this information for reasonable project typologization in the process of expert panel discussion. The systematized projects' descriptions and classifications can be found in the supplementary materials [69].

We assigned the supported innovation projects to the nine categories based on the conceptual design of the typological analysis (see 2.1. Conceptual design of typological analysis), and using the expert panel method. Being interdisciplinary experts in the innovation sphere, we discussed each project through the lens of the suggested typological analysis and synthesized our arguments and views to identify the types of innovation project effects.

The supported innovation projects were classified by two main categories within our typological matrix: SD ethical dimensions and the scale of SD effects (Table 2). We must note that supported innovation projects can have more than one SD effect according to the suggested typology. For example, the project "Non-aerodrome aircraft «BELLA»" has three multi-level effects: operative environmental effect (as it uses energy-saving technologies in one particular vehicle), tactical social effect (as it is a new technology of increasing transport possibilities and approachability of remote and under-populated territories), and tactical economic effect (as it is a new cost-efficient technology for delivering people and loads to non-aerodrome places). According to this calculation technique, 31 supported innovation projects during 2009–2015 had 50 topological effects, and 66 supported innovation projects during 2016–2017 had 86 topological effects. The calculations are available in the supplementary materials [69].

The results of the typological analysis of innovation activities (IA) are formalized in a matrix form (1).

$$\text{IA (SD)} = \begin{bmatrix} SEnI & SSI & SEI \\ TEnI & TSI & TEI \\ OEnI & OSI & OEI \end{bmatrix} \tag{1}$$

The typological analysis involved the following steps:

(1)　Classification of supported innovation projects according to their SD effects. Each project was allocated to the appropriate classification group according to their potential SD effects. An innovation project could be allocated to several classification groups if it implied different effects simultaneously, for example strategic environmental and tactical economic effects, or tactical social and operative environmental effects. Calculation of the number of innovation projects in each cell of the typological matrix for classification of sustainability-oriented innovation activities in two periods: 2009–2015 and 2016–2017 (Table 3).

(2)　Formalization of the typological analysis results in matrices consisting of absolute and relative values and visualization of the results.

(3)　Identification of supported innovation projects belonging to mining operations and the oil and gas industry.

(4)　Assessment of basic conditions for an effective sustainability-oriented innovation support system on the basis of typological analysis, including comparison of typological matrices by time periods (2009–2015 and 2016–2017).

(5)　Development of recommendations on which transformations in decision making policy are necessary to significantly increase the innovation system's potential impact on achieving wider SD goals.

## 3. Results

The situation of compliance of the Tyumen innovation support infrastructure with SD ethics is presented in Table 3 and formalized in matrices consisting of absolute values (Table 2, Equations (2) and (3)) and relative values (Equations (4) and (5)).

**Table 3.** Typological analysis of supported innovation projects [69].

| SD Ethics Dimensions | The Scale of SD Innovation Effects | | | | | |
|---|---|---|---|---|---|---|
| | Environmental | | Social | | Economic | |
| | 2009–2015 | 2016–2017 | 2009–2015 | 2016–2017 | 2009–2015 | 2016–2017 |
| Strategic | 2 | 0 | 0 | 1 | 2 | 1 |
| Tactical | 13 | 4 | 2 | 8 | 14 | 9 |
| Operative | 2 | 9 | 1 | 15 | 14 | 39 |

During 2009–2015, most attention was paid to innovations with implicit tactical and operative economic effects and tactical environmental effects—from 31 innovation projects in total, 14 showed mid-term economic effects, 14 showed short-term economic effects, and 13 showed mid-term environmental effects. During 2016–2017, among 66 supported innovation projects, 49 had an economic impact, 24 had a social impact, and only 13 had an environmental impact. Economic innovations mostly were focused on short-term economic returns (39); some considered mid-term economic effects (9), and only one project implied strategic economic effects.

$$\text{IA } (2009 - 2015) \ = \ \begin{bmatrix} 2 & 0 & 0 \\ 13 & 2 & 14 \\ 2 & 1 & 14 \end{bmatrix} \tag{2}$$

$$\text{IA } (2016 - 2017) \ = \ \begin{bmatrix} 0 & 1 & 1 \\ 4 & 8 & 9 \\ 9 & 15 & 39 \end{bmatrix} \tag{3}$$

$$\text{IA } (2009 - 2015) \ = \ \begin{bmatrix} 4\% & 0\% & 4\% \\ 26\% & 4\% & 28\% \\ 4\% & 2\% & 28\% \end{bmatrix} \tag{4}$$

$$\text{IA } (2016 - 2017) \ = \ \begin{bmatrix} 0\% & 1\% & 1\% \\ 5\% & 9\% & 10\% \\ 10\% & 17\% & 45\% \end{bmatrix} \tag{5}$$

In both periods, innovations providing strategic SD effects comprised a very insignificant share of the supported pool of innovations: 8% during 2009–2015 and only 2% during 2016–2017. During 2009–2015, most of the innovation effects (58%) were at the tactical level of solving SD problems. During 2016–2017, tactical innovations decreased to 24%. During 2016–2017, the focus of innovation support policy noticeably shifted to operative level innovation effects—the share of operative innovations was 72%, in comparison with 34% in the previous period.

Again, according to the matrices above (Equations (2)–(5)), the main tendency of the regional innovation support policy was strengthening the focus on short-term regional prosperity during 2016–2017 in comparison with 2009–2015. The shares of innovation projects affecting social and environmental spheres were redistributed: the share of projects with environmental effects decreased, and the share of projects with social effects increased during 2016–2017, but their total share stayed in the minority—less than 50%.

During 2009–2015, 45% of the supported innovation projects were mining operations and oil and gas extraction, and 39% of innovation projects were in both the mining industry combined with other segments (production, services, machine building, and construction). Only 16% of the innovation projects examined focused on other economic spheres outside the region's mining operations and oil and gas extraction activities. During 2016–2017, the share of innovations focusing on mining operations and oil and gas extraction indicated an overall increase of project support to 67% of the total which the

technopark took under its wing. Simultaneously, the amount of innovation projects focusing on the effects of other industries decreased by 24%.

## 4. Discussion

Analysis of basic conditions for effective sustainability-oriented innovation activities revealed an increasingly reactive and timeserving character of the Tyumen innovation support policy. The majority of supported innovation projects served mid-term and short-term regional sustainability goals. A focus on quick investment returns, maximization of profit, and commercialization in innovation support decision-making determined the innovation support policy in the Tyumen region, while systemic SD effects (ecological, economic, and social) were given minor consideration.

In addition, this expedient approach to sustainable development has become entrenched over time. The total share of innovation projects characterized by operative effects grew from 34% (2009–2015) to 72% (2016–2017). These results reflect prevailing shot-termism in innovation policy decision-making [43,58], and serve to demonstrate the general unsoundness of today's innovation efforts initially aimed at SD goals. The employed typological analysis allowed the research to reveal the reason for this "unsoundness" at the level of basic conditions—the lack of marked focus on systemic and long-term SD effects. The model also allowed us to explain the extent to which existing research and development and national systems of innovation frameworks for science, technology, and innovation policy are currently unfit for addressing longer-term environmental and social challenges [21].

The findings indicate that sponsored innovation projects primarily served the goal of economic growth at the mid-term and short-term levels. This particular case reflects continuing domination of economic growth ideas in social development [1,15]. Environmental innovations received government support, but they also focused on the tactical level, i.e., technologies for waste-free or resource-saving production, and technologies preventing pollution and other environmental threats in mining industries, without a radical shift to developing technologies and innovations in non-mining industries. During 2016–2017, environmental innovations had the least share of support and achieved environmental impact only at the operational level. However, this is not enough strategic effort to making necessary major changes toward a sustainable level of depletion of nature [4,26].

During 2009–2015, the least attention was paid to innovations targeting the social sphere: only three projects dedicated to improving human safety and automation of manual operations were supported. During 2016–2017, the number of projects aimed at the social aspect of sustainability substantially increased (from 3 up to 24), but the focus remained on solving social problems at the operative level [70]. Despite the increase in the number of projects, we cannot assert that innovation support policy really has an effect on common prosperity and long-term vitality. Strategic social innovations providing radical political changes and transformation of social behavior are crucial for overcoming system collapse [26,71,72].

The typological analysis indicates the evident focus of Tyumen innovation support policy on solving the SD problems primarily at tactical and operative levels. Supported innovation projects focused mostly on regional short-term prosperity and economic growth [58,73]. In other words, the focus of regional innovation support policy remained on achieving economic effects without radical societal transition to a sustainable development path. Supported innovation projects focused mainly on "curing symptoms" of regional economic, social, or environmental problems, but not on systemically eliminating their fundamental causes.

The problem is not in the lack of peoples' inventiveness or talents [74]. Rather, the problem is how to mobilize these assets [75], or to be more precise, of how to manage already existing innovative potential. "There are promising new technologies with better environmental performance. But many of these new technologies are not (yet) taken up" [76]. This is partly related to economic reasons, but also to social, cultural, infrastructural, and regulatory reasons [76]. Considering the possibility of supporting existing innovations with greater SD potential and the potential of innovation support policy to foster innovation with deeper and more scalable SD effects, we should begin to solve this

problem with a consistent methodological background for decision-making in the sphere of innovation support. Developing and implementing strategic systemic innovation requires a different approach to making decisions about supporting innovations; one based on a systematic, long term vision [21,65,66]. This change requires an updated toolbox for decision-making support, such as the typological analysis presented in this work. Theoretically, developing and testing tools, increasing the impact of innovation activities on achieving SD goals, and complementing modern research in the sphere of SD societal transition will strengthen them practically and methodically [40,43,70,76,77].

Until 2015, the Tyumen regional innovation support policy was mainly directed at supporting mining industries (Figure 1). During 2016–2017, the priorities of the innovation support policy shifted from mining industries to other ones. The share of supported innovations that targeted non-mining operations or oil and gas extraction increased from 39% during 2009–2015 to 67% during 2016–2017. This situation corresponds to the traditional structure of the Tyumen regional economy, which is mainly based on oil and gas production and has its constructive and destructive side. There are some objective stabilizing (conservative) forces providing current system stability that are reflected in Geel's notion that, "existing systems are "locked in" at multiple dimensions, they are stable and not easy to change" [76]. In addition, long-term system vitality may require a major transition to different structures and conditions, eliminating accumulated systemic problems. From this point of view, the current system stability, based on a resource-oriented economy, obviously hampers the region's deep transition to SD and a green economy. Regional elites play a major role in regional innovation policy. Regional innovation policy "as usual", or a resource-oriented economy, is supported mostly by the interests of particular market agents (such as oil and gas companies) [72], with prejudice to the interests of the long-term vitality of the regional socio–economic system and its sustainable development over the long-term. As Motesharrei et al. noticed, a buffer of wealth allows "elites to continue "business as usual" despite the impending catastrophe" [26].

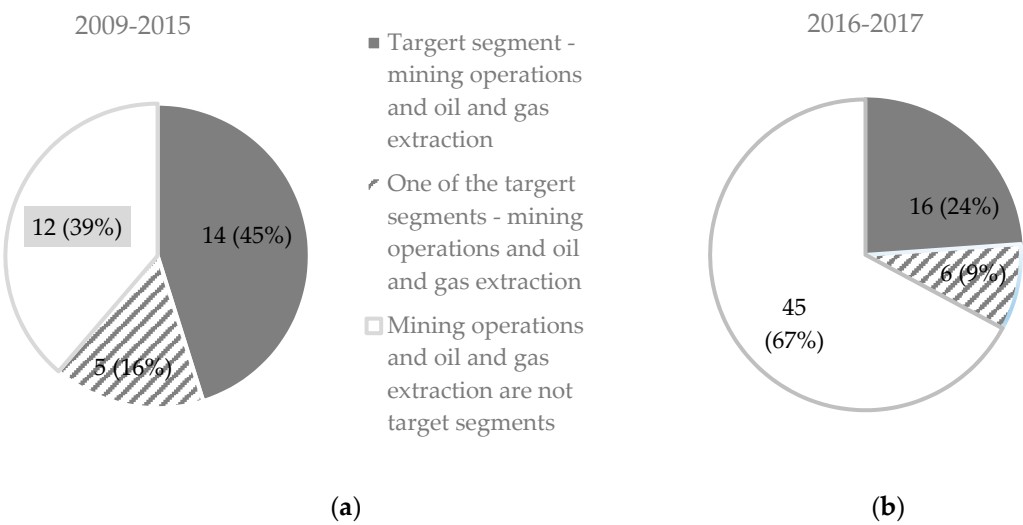

**Figure 1.** Classification of innovation projects supported by Tyumen Technopark by industry (their number and share of total, %): (**a**) 2009–2015; (**b**) 2016–2017 [69].

There is still a strong need in the system for socio–technological innovations and to change consumer culture in general to provide SD societal transition. This requires strong governmental support of strategic environmental, social, and economic innovations that sufficiently decrease the depletion of natural resources, replacing unclean and unsafe technologies, production, and industries with totally clean and safe ones, and favoring changing social–economic relationships into more fair, constructive, and sustainable ones in the long-term.

## 5. Conclusions

### 5.1. Recommendations

The crucial role of social innovations in achieving SD goals should be understood and accepted by society and by a government. Specifically, primary support should be provided to the innovations that are able to transform basic mechanisms of the current social–economic system, in which global problems continue to be generated. Strategic social innovations need to change the general culture of disunity, consumption, and competitiveness into humane and collaborative ones.

The suggested typological analysis technique can be used as a highly-needed "simple stabilizing feedback mechanism" [1] that facilitates the sustainable development of a complicated social system. Implementing the analysis technique in the decision-making processes would help change social behavior and expectations, and facilitate institutional change regarding norms, standards, and regulations [34]. Involving people in the processes of innovation support decision-making, monitoring the results of implementing innovation, and estimating them in accordance with strategic SD goals are the essential factors for a socio–technical transition towards long-term sustainability.

In a period of deep social transition, innovation activity acts as an adaptive mechanism, transforming the society from one quality statement to another. In this period of transformation, the quality rather than quantity of innovation activities is important. To increase the impact of innovation activities in achieving SD goals innovation decision-making policy development and support should be designed in accordance with basic principles: (1) the accordance of innovation activities with sustainable development ethics, and (2) their marked focus on systemic and long-term SD effects. Prioritization of supporting SD-oriented strategic innovation will provide a successful transition to the next level of regional sustainability and will provide new frameworks for other types of innovations.

### 5.2. Research Implications

Research results impact and complement the existing research on estimating innovation activities' effectiveness in the context of SD societal transition. The suggested typological analysis technique might be integrated into the complex assessment of the innovation system's quality and the innovation policies' results. It aims to increase the effectiveness of innovation support institutes, programs of innovation development, and innovation systems on the global, national, regional, local, and organizational levels. The typological analysis provides information for redesigning innovation systems, innovation policies, and decision-making policies through different bodies of innovation infrastructure, increasing their impact on achieving SD goals.

As Van der Vleuten noticed, "Deep Transitions research is timely and urgent; however, in sustainability transition studies, individual systems remain the dominant unit of analysis" [70]. However, the suggested typological analysis technique does not have conceptual limitations in its application in the analysis and redesign of overall world innovation policy and development, despite the fact that it was tested on the example of a particular regional innovation support system.

As was noticed by Cancino et al., partial theoretical perspectives and experiences of innovation effects lead to "significant oversight of their potential and limitations" [50]. This research adds to the theoretical base of innovation assessment and decision-making from a SD perspective, suggesting a new typological matrix classification for innovation and an appropriate typological analysis technique that can be used both in theoretical research (e.g., in an ontological analysis of innovation systems and policies) and in practical analysis and decision-making.

### 5.3. Limitations and Further Research Development

In this research, the typological analysis was applied to one main object of the regional innovation support system. In further research this method can be extended to: (1) all objects and programs of an innovation support system (grants for innovative projects, governmental support programs for small

and medium enterprises, innovation competitions, etc.); (2) all elements of the regional innovation system (science, education, innovation policy, social innovations); and (3) different levels—local, organizational, regional, national, or global. It is felt that this method will be useful for both the analysis and redesign of innovation activities (systems, policies) in the most effective way to achieve sufficient SD social transformation.

Nevertheless, the results of the typological analysis of Tyumen Technopark innovation support activities were quite representative, because Technopark is the main institute of the Tyumen innovation support infrastructure and the only one that provides systemic government support of regional innovations at all stages.

One of the research limitations is analyzing data in unequal time periods. The reason for that was a significant difference in the number of supported projects in the Technopark's development period during 2005 till 2016 (31 projects) and the last two years of its work in the period 2016–2017 (66 projects). These pools of supported innovation were analyzed and compared to find if there were trends regarding changing the focus of innovation support policy in the last years. The analysis of time series data can be done annually if the work of the infrastructure object is quite stable in the number of supported innovation projects.

Typological analysis of the basic conditions for effective sustainability-oriented innovation activities allowed us to grasp the character of innovation support policy and develop appropriate recommendations. In this research, a number of supported innovation projects were used as elements of the typological matrix because of their data relevance and availability. However, the procedure of the suggested typological analysis can be modified and other indicators can be used in the typological matrix if necessary data are available, such as the budget of each project, size of governmental support, or monetary equivalent of the SD effect. This will allow researchers to reveal more specific details in the analysis of basic conditions for the effective sustainability-oriented innovation activities, and allow them to extend and clarify recommendations for redesigning decision-making policies.

In the framework of the article, we described only one case for application of the suggested method, but we hope for its wide application in further innovation. The suggested typology will be helpful for innovators, government, investors, and society for the purposes of making SD-oriented decisions and matching interests of different groups in the innovation sphere.

**Supplementary Materials:** The following are available online at http://dx.doi.org/10.17632/3c95r56fjg.1.

**Author Contributions:** A.O.L. developed the main conceptual ideas, designed the methodological framework and the typological model, processed data, and performed the initial analysis. D.L.D. contributed to the analysis of the background data, refined the theoretical basis of the typological model, added to the data analysis and recommendations, and reviewed the paper. O.V.Z. contributed to the analysis of the context and background, formulated research limitations, and made a draft preparation. Y.K. contributed to interpretation of the results and developed suggestions for further research. All authors provided critical feedback and helped structure the manuscript and contributed to the editing of the final version of the manuscript.

**Funding:** This research was funded by the Russian Science Foundation (Project No. 14-38-00009). The program targeted management of the Russian Arctic zone development with the Great St. Petersburg Polytechnic University and The Russian Science Foundation, 109992, Russian Federation, Moscow, http://rscf.ru/. The study did not get external financial support.

**Acknowledgments:** The article is prepared in the framework of the research associations of "Innovation development of Russian Arctic regions and economic sectors" and "National Arctic Scientific-Educational Consortium". We are thankful to the team of the Center of Academic Writing of Tyumen State University, and Zhuravleva Nadezhda and Valeria Evdash who lent professional support in preparing this manuscript.

**Conflicts of Interest:** The authors declare no conflict of interest.

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
