# Peer review of "Managing Innovation Resources in Accordance with Sustainable Development Ethics: Typological Analysis"

_resources, doi:10.3390/resources8020082_

Round 1

Reviewer 1 Report

The current manuscript suggests an interesting topology of sustainability-oriented innovation projects. It was built upon two solid axes: 1) three dimensions of triple bottom line framework (environmental, social, and economic) and 2) three widely-used project scales (strategic, tactical, and operative). However, the current level of scientific investigation does not really match the standards of Resources. This will be discussed in the points below.

Major issues:

Lines 139-142: This argument extends too far. A single case study can be theorized (and therefore generalizable) when it represents an extreme, unique, or revelatory case (Yin, 1994). Please corroborate your argument or note this as an additional limitation in Section 5.3.

Lines 324 and 331: I see the authors analyzed the data in 2 separate periods (i.e., 2009-2015 and 2016-2017) in accordance with the data sources. Although those two periods demonstrate the changing attention of SD projects from tactic to operative, both are from Tyumen regional government (plus, it is not clear what “… additional projects descriptions in the Internet …” in line 311). In addition, how did you deal with the projects going over those two periods?

Table 3 and its explanation in lines 340-346 do not match at all. For instance, I added up the numbers in Table 3 and got 50 (= 17 environmental + 3 social + 30 economic) for 2009-2015 and 86 (= 13 environmental + 24 social + 49 economic) for 2016-2017. Please explain how your samples could be 31 and 66, respectively. If there were multi-purpose projects, more details of them should be provided rather than overstating the sample size by counting such project twice or three times.

My biggest concern on this manuscript is the absence of its analytical rigor and corresponding contribution for academicians or practitioners.

o   Section 3 (“Results”) is a simple description of their primary data based on the provided 3-by-3 matrix. Figures 1 and 2 are simple repetitions of lines 347-348.

o   It is not clear how the authors classified the primary data to each of the nine categories in Section 3 (“Results”). I do not think Tyumen regional government used the classification of strategic/tactic/operative, then how did you deal with the projects with characteristics of more than one types?

o   More importantly, how did you confirm the differences among the types? Wasn’t event simple one- or two-way ANOVA doable for the two time periods used or among 2-3 biggest categories (e.g., 2009-2015 TEnI vs. TEI vs. OEI or 2009-2015 OEI vs. 2016-2017 OEI)?

o In a current format, therefore, it is unclear what readers could learn other than what Tyumen regional government has been doing for sustainable innovation since 2009 from this article. There are very few generalizable or widely applicable findings/cases provided.

Minor comments:

Section 1.2 and section 1.3 titles are identical (“Focusing on long-term rationality”). This must be a typo.

Lines 49-102: What do you mean by “our prior investigations”? Does this manuscript have any preceding studies?

“Deep Transition” in line 116: Why is this title cased? Put a quotation mark and add a citation(s) if it is coined by someone else.

Drop “on” in line 134.

Line 158: Do not quote a title – italicize it.

Line 507: Typo – “… since 2005 till 20016”

I appreciate the opportunity to review the paper. Best wishes to the authors for future endeavor.

Reference:

Yin, R.K. (1994) “Case Study Research: Design and Methods” (2nd Ed), Sage Publications, Thousand Oaks, CA.

Author Response

Our great thanks to Reviewer 1 for very attentive reading and thoughtful remarks and also for noting typos and mistakes which me unfortunately missed.

Reviewer 2 Report

Dear colleagues,

Thank you for your research paper which is of interest and importance due to the acute point of tourism sustainability. The article does not arouse any objections and could be published as it is.

Kind regards,

Author Response

We would like to thank the Reviewer 2 for the time contribution and the chance to publish the article and thus bring acute SD issues to the scientific debates.

Reviewer 3 Report

The research topic appears to be worthwhile and might contribute to the scientific debate.

Keywords - could be more relevant to the topic.

Article structure - not clear why Introduction includes literature review; also 1.2 and 1.3 have the same title.

Concepts- why is the term SD ethics used for SD triple dimensions? (seems inappropriately used).

Methodology -  why is the analysis conducted for the two unbalanced periods (7 years vs. 2 years); might lead to misleading results.

Results and discussion: interesting results and conclusion, well pointed out limitations.

References - a very, very large number of references (still, only one from Resources); however, I am not sure that the correspondence between the text and the cited references is always correct (e.g. ref 1 should rather be ref 14 and there are more other situations); the order of the references in the text seems unclear (e.g. after ref 63 - line 288 follows ref 73-77 -line 299), too many references do not comply with the journal style.

There are some spelling mistakes (e.g. line 214, line 409).

Author Response

We thank the Reviewer for the time contribution and good points for improving the article. We were glad to work over the remarks and we are pleased the results of our research were highly evaluated.

Reviewer 4 Report

Please see attached my comments

Author Response

Remark 1. Websites (line 307) should be cited and moved to reference list

Revision 1. Thank you for your comment. We added 3 websites in the reference list (66-68).

Remark 2. Figures must have sources

Revision 2. Thank you for your note. We added sources in Table 3 and Figures.

Remark 3. A paragraph with the structure of the paper must be added at the end of the Introductory part.

Revision 3. Thank you for your suggestion. We added a paragraph with the structure at the end of the Introductory part. (Lines 148-152).

Remark 3. Reference list must be reviewed. The year of the journal must be bolded. Here is an example: 1. Author 1, A.B.; Author 2, C.D. Title of the article. Abbreviated Journal Name Year, Volume, page range.

Revision 4. Thank you for your remark. We revised references and corrected mistakes.

In conclusion, we thank the Reviewer for the time contribution and good points for improving the article. We were glad to work over the remarks and we are pleased the results of our research were rather highly evaluated.

Round 2

Reviewer 1 Report

I would like to commend the authors on their tremendous efforts to improve the manuscript. They have well addressed my comments EXCEPT the following two that need a bit more efforts:

Remark 1.4: The projects I was referring to here were the one running across both periods, for instance, the project implemented from 2015 to 2016. Please explain more about how you counted those cases if there was any.

Remark 1.6: I understand your point but still believe you do not need two separate figures. Please delete either Figure 1 or Figure 2 and add percentage or counts in parentheses.

Thanks.

Author Response

Remark 1.4: The projects I was referring to here were the one running across both periods, for instance, the project implemented from 2015 to 2016. Please explain more about how you counted those cases if there was any.

Comment 1.4: Thank you for clarifying your remark. It is really important to clear up this moment in calculations.

In our calculations, we considered innovation projects only in the year of the Technopark’s decision to support them. According to our research purpose (“The practical purpose of this research is to provide respective audiences with a relevant tool for assessment and conceptual design of innovation activities and innovation systems tuned to provide significant SD social transformations”) it made sense to consider and analyze the characteristics of the proposed effects of innovation projects just in the year of a decision making. Transformation power of the suggested analytical tool and its influence on the conceptual design on innovation activities implies understanding the content of the projects in the moment of decision making. Following your remark, we decided to add some comments on this this issue in the text.

Revision 1.4: We clarify this issue adding the text in Lines 322-325: “In our calculations, we considered innovation projects in the year of the Technopark’s decision to support them. For example, if the project was supported in 2015 it was considered only in this year (2015) in spite of its further developing in 2016 and later”.

Remark 1.6: I understand your point but still believe you do not need two separate figures. Please delete either Figure 1 or Figure 2 and add percentage or counts in parentheses.

Revision 1.6: We deleted the figure 1 as it was less valuable and mainly repeated the results from the Table 3 and matrixes 1-5. And we saved Figure 2 as it had some new focus on data.

Revision 1.6: According to the change above we made minor changes in the text with citations on Figure 1 and 2 and corrected the sentence in the Line 388: “Again, according to the matrixes above (2-5)…

Revision 1.6: We parenthesized percentage in Figure 1 (Line 394).

In conclusion, we want to thank one more time the Reviewer 1 for helping us refining the article and giving highly valuable advices. We hope that these common efforts and time investment will bring the benefit serving for the common prosperity through better understanding how to move to sustainable development.
